# An Analysis of the Reliability of a Bus Safety Structure on Carrying Out the Numerical and Experimental Tests

**DOI:** 10.3390/s20247092

**Published:** 2020-12-11

**Authors:** Tautvydas Pravilonis, Valdas Eidukynas, Edgar Sokolovskij

**Affiliations:** 1Department of Automobile Engineering, Transport Engineering Faculty, Vilnius Gediminas Technical University, J. Basanavičiaus g. 28, LT-03224 Vilnius, Lithuania; edgar.sokolovskij@vgtu.lt; 2Department of Mechanical Engineering, Faculty of Mechanical Engineering and Design, Kaunas University of Technology, Studentų st. 56-344, LT-51424 Kaunas, Lithuania; valdas.eidukynas@ktu.lt

**Keywords:** safety, reliability, structure, modeling, finite element method, bus

## Abstract

In the paper, the reliability of a spatial tubular structure of a bus safety frame formed of different steel profiles is discussed. A methodology for the bus safety structure modeling is presented herein by applying numerical methods that enable us to simulate virtually a test for assessing bus rollover crashworthiness according to the United Nations Economic Commission for Europe (UNECE) Regulation No. 66, and also to assess and ensure the reliability and safety of the structure under operating conditions. The simulation has been performed by applying the mixed method of kinematical analysis and finite elements. In the course of the calculations, physical and geometrical non-linearity of materials was assessed. In addition, an experimental rollover test according to UNECE Regulation No. 66 was performed in this work, striving to verify the provided methodology for modeling by applying numerical methods. For the experiment, an identical safety structure and a rollover stand (identical to the one used in modeling) were used. The rollover test was shot by a Phantom v711 high-speed camera. In the paper, the results of kinematical and dynamic analysis from applying the finite element method and the ones of the experimental test, as well as their comparisons, are provided. It is assessed whether the developed safety structure model is reliable and suitable for use.

## 1. Introduction

Buses are very popular vehicles. Medium-size buses for transportation of passengers become particularly popular because they provide more freedom, both for the carriers and the passengers; in addition, they are environmentally friendly and economical.

However, the growing number of used buses is accompanied by the growing of traffic events where these vehicles are involved. Among all the traffic events caused by buses, an event related to bus rollover causes the maximum damage because it may cause even several fatalities per event. As compared to rollover of a car, a bus rollover occurs rarely; however, it is much more dangerous because of a larger number of passengers transported [1,2,3,4,5].

In 2016, a total of 594 persons perished in traffic events caused by buses [6]. Thus, in the course of designing bus safety structures, their reliability in normal and critical conditions should be assessed and ensured.

Medium-size buses for transportation of passengers, as distinct from large-size ones usable for the same purpose, are manufactured in two stages and they consist of two principal elements: (a) a chassis and (b) a passenger compartment. In the first stage, a chassis, an engine, and a transmission are manufactured on a conveyor. In the second stage, a passenger compartment is designed according to an individual order and then equipped on the chassis made in the first stage. In the second stage, safety of the manufactured vehicle and safety of traffic participants, passengers, and other persons, as well as satisfaction of the set safety requirements, and the growing needs of the community are considered [4,7,8,9].

According to Chirwa et al. [10], a rollover of a bus is the most dangerous when the bus falls from a certain height onto a solid surface. Such a rollover most frequently occurs when motion of the bus is curvilinear, and it is affected by centrifugal acceleration [11].

A rollover of a bus is very dangerous because it causes a considerable deformation of the roofing and sidewalls of the bus. In the case of such a deformation, the residual space is reduced dramatically, and the passengers can be injured or even wounded resulting in death. Thus, it should be emphasized once more that reliability of the bus safety structure directly impacts safety of the passengers.

A dynamic bus rollover test is one of the principal requirements set for medium-size buses. It is described in United Nations Economic Commission for Europe (UNECE) Regulation No. 66 [11,12].

By the way, for assessing suitability of a bus structure, also another methodology can be applied—it is a quasimetric bus roofing load test, according to the Federal Motor Vehicle Safety Standard 220 (FMVSS 220) [9,13]. However, a comparison of the results obtained upon applying the UNECE Regulation No. 66 and FMVSS 220 carried out by Liang and Nam [13] shows that the passenger compartment and the residual space were more affected and a higher risk was caused for the passengers in the case of the bus rollover test according to UNECE Regulation No. 66. In addition, it was found that the structure that conforms to provisions of FMVSS 220 does not conform to provisions of the UNECE Regulation No. 66 [7,9,13]. Thus, in this paper, the strength of the bus structure is analyzed upon applying a test on a bus rollover according to the United Nations Economic Commission for Europe (UNECE) Regulation No. 66.

The said Regulation provides that a structure of a vehicle should remain free of any violations of the residual space of the vehicle during a coach rollover test and after it [14].

The consequences of a rollover depend on the velocity of the rollover, the height of the center of gravity, and the distance to the surface where the bus falls [15].

In a bus rollover, the passengers can fully or partially fall out from the bus or be pressed down inside the vehicle and thus be wounded, causing death. At present, the United Nations Economic Commission for Europe (UNECE) Regulation No. 66 is considered a principal standard for bus rollover crashworthiness assessment tests, so all bus manufacturers are obliged to observe its provisions in real tests or to assess reliability and strength of the structure, as well as its resistance to a rollover, by applying the computer-aided modeling [3,16].

Therefore, the structure of buses should strictly conform to provisions of the United Nations Economic Commission for Europe (UNECE) Regulation No. 66.

Various tests, as well as a selection of relevant systems and operations or algorithms, are very expensive. For this reason, a majority of manufacturers apply computer-aided simulations [17,18]. This dynamic rollover test is not an exception, because it is sufficiently costly and complicated—it requires a prepared vehicle and a special stand for rollover tests [9,19]. Thus, a majority of manufacturers prefer to apply the finite element method for assessing the conformity of the structure to provisions of the United Nations Economic Commission for Europe (UNECE) Regulation No. 66 [4,9,11,19,20].

## 2. Methodology

The methodology specifies the requirements and conditions to be met during the rollover test of the bus safety frame. A description of the object in question is also provided.

### 2.1. The Object under Research

The object under research was a spatial (consisting of elements that are not situated on the same plane) tubular structure of a constant shape (the distances and the angles between all the points of its elements are constant) formed of different steel profiles. For the principal safety arches, 80 × 40 × 3 mm profiles were used. The safety arches were interconnected by 40 × 40 × 2 mm profiles. In the structure, 30 × 30 × 2 and 40 × 30 × 2 profiles were used as well. Stiff edges and V-shaped hinges were also used in the safety frame. All the profiles were interconnected by welding. 

The mass of the object under discussion (with a supplemental mass) was 610 kg. The length of the object was 4050 mm, its width was 2400 mm, and its height was 2557 mm.

### 2.2. The Rollover Test

The test stand consists of a tip-up platform and a pit with a solid surface (or a tip-up platform on a height of 800 mm over a solid surface). The tested vehicle should be put on the tip-up platform.

The principal requirement to be satisfied by the vehicle under testing is the position of the center of gravity—it should coincide with the one of a fully equipped vehicle. In addition, the total mass of the vehicle and the distribution of masses in the vehicle under testing should fully conform to the total mass of a fully equipped vehicle and the distribution of masses in it. 

While carrying out a rollover test, the platform under the vehicle should be uniformly lifted until the vehicle achieves an unstable position and a rollover starts. The angular velocity of the tilt platform should not exceed 0.087 rad/s. The surface which the vehicle falls on should be dry and smooth [18].

Simulation of the bus safety structure rollover was performed, striving to assess a deformation of the structure (the velocity and the acceleration of the upper angle of the structure when it strikes against the concrete surface). A computational model of the bus safety structure rollover is shown in Figure 1.

It goes without saying that a rollover test of a fully equipped vehicle is the most reliable method, and the obtained results are the most accurate; however, such tests are very expensive. Taking into account this circumstance, other methods of equal value are applied. One of the methods is a rollover test using sections of the safety frame. In this case, the results will not be highly accurate because they reflect the strength of the chosen section (a part of the frame) only. Usually, the values of rigidity of different sections of a bus differ. In addition, the rigidity of the structure of interconnected sections is considerably higher, as compared to the rigidity of an individual section [14].

Another rollover method of equal value is the computer-aided rollover modeling method [14].

As it was mentioned above, a safe residual space should remain for the passengers after the rollover test. The said space can be described as follows—it is a space for passengers, the crew, and the driver, not usable for any other purpose, that ensures a high probability of avoiding loss of life in case of the bus rollover.

On a rollover test of the load-carrying body structure, the residual space shown in Figure 2 should not be violated [14].

In order to verify the presented calculation methodology, the results of the rollover experimental test and the computer-aided analysis are compared in this work.

### 2.3. The Energy Calculation

The energy reference value of the safety frame should be calculated according to the United Nations Economic Commission for Europe (UNECE) Regulation No. 66 “Uniform Technical Prescriptions Concerning the Approval of Large Passenger Vehicles with Regard to the Strength of Their Superstructure”, as follows the value of reference energy (*E_R_*), which is the product of the vehicle mass (M), the gravity constant (g), and the height (h_1_) of center of gravity with the vehicle in its unstable equilibrium position when starting the rollover test, as shown in Figure 1 [14]:(1)$$E_{R}=M\cdot g\cdot h_{1}=M\cdot g\;\left[0.8+\sqrt{h_{0}^{2}+{\left(B\pm t\right)}^{2}}\right],$$
where M = M_k_, the unladen kerb mass of the vehicle type if there are no occupant restraints, or, M_t_, total effective vehicle mass when occupant restraints are fitted, and M_t_ = M_k_ + k M_m_, where k = 0.5 and M_m_ the total mass of the restrained occupants, *h*_0_ the height (in meters) of the vehicle center of gravity for the value of mass (M) chosen, t perpendicular distance (in meters) of the vehicle center of gravity from its longitudinal vertical central plane, *B* perpendicular distance (in meters) of the vehicle’s longitudinal vertical central plane to the axis of rotation in the rollover test, g gravitational constant, h_1_ the height (in meters) of the vehicle center of gravity in its starting, unstable position related to the horizontal lower plane of the ditch [14].

## 3. Results

### 3.1. The Numerical Analysis of the Safety Structure

The calculation consists of two stages. In the first stage, the kinematical calculations for establishing the tilting angle when the structure becomes unstable, the time until it is falling onto the concrete surface, the angle of the impact of the frame structure against the concrete surface, and the angular velocity of the structure around the rollover axis at the moment of the impact were carried out. In the second stage, calculations of the stressed state of the deformed structure at the moment of the impact upon marginal conditions of the task using data from the results of the kinematical analysis were carried out. In this way, dynamic analysis of the safety frame was performed using realistic model boundary conditions.

The kinematical calculations were carried out by applying the annex Motion Dassault systems of SolidWorks (SolidWorks 2019 version). SolidWorks Motion is the light version of MSC ADAMS, as the world’s most famous and widely used Multibody Dynamics (MBD) software. Multibody Dynamics simulations are a powerful method to study both the kinematic and the dynamic behavior of complex systems. MBD simulations are suited to study the dynamic behavior of interconnected rigid and/or flexible bodies undergoing large translational or rotational displacements. The motion of those bodies is calculated based on applied loads and boundary conditions defined. Typically, these simulations have short calculation times, making them the preferred tool to conduct parameter studies or optimizations very efficiently.

The analysis of the impact was performed by applying the Ansys Autodyn system (Ansys Autodyn 2019R1 version) from ANSYS Inc. Ansys Autodyn simulates the response of materials to short duration severe loadings from impact, high pressure, or explosions. It is best suited for simulating large material deformation or failure. Complex physical phenomena, such as the interaction of liquids, solids, and gases, can be modeled within Autodyn. Integrated within Ansys Workbench with its own native user interface, this program has for decades led the industry in ease of use, enabling consumers to get accurate results with the least amount of time and effort.

A spatial geometrical model of the safety structure for the verification is presented in Figure 3 below. In the calculations, several computational models were formed. The model for kinematic analysis was a spatial tubular structure of a bus safety frame, on the basis of which a natural safety frame was also produced. For dynamic (impact) analysis, a shell model was developed, which allows the significant reduction of the number of finite elements, thus shortening the computation time. 

The natural bus safety frame fragment was welded using square and rectangular cross-section profiles. The thicknesses of these profiles were 2 and 3 mm. In addition, stiffening edges and V-shaped hinges were used for welding the frame. The thicknesses of these elements were 1.5, 3, and 4 mm. The computational model for kinematic analysis was created using a direct spatial model of the frame, on the basis of which the frame was designed for a natural experiment, that is, the geometry of the model corresponds exactly to the real frame model. All frame elements were fixed to each other. Distance sensors were modeled at three corners of the model frame to accurately determine the moment of contact. The frame rollover computational model is presented in Figure 4 below. Here, 1—the concrete surface (base); 2—the tip-up platform; 3—the tip-up spatial model of the safety structure with an auxiliary stand; the width of the stand conforms to the width of the rear axle of the coach and the lower part conforms to the point of a wheel’s support on the base; 4—an additional mass of 610 kg; and 5—sensors for contact detection.

In Figure 5, the initial position of a rollover is shown. It was established by calculations on varying the angle until the structure of the frame loses its stability and starts turning over. To determine this initial angle, the position of the center of gravity of the frame structure was calculated and the frame was tilted until this center of gravity was extended beyond the axis of the tip-up platform hinge (about which tip-up was made).

In course of the kinematical analysis, it was found that the initial angle when the structure of the frame (loaded with an additional mass of 610 kg) loses its stability and starts turning equals 40 degrees. At the said moment, the velocity and the angular velocity of the top point of the frame equals 0. An additional mass of 610 kg was loaded to bring the frame as close as possible to real operating conditions (simulating seats and passengers). The double profile was mounted at a height to match the center of gravity of the passenger bus.

In the course of the analysis of the frame rollover, it was found that the top point of the frame structure reached the base within 1.64 s. The velocity of the angle at the moment of the impact against the base was about 5.97 m/s. In Figure 6, the alteration of the angular velocity of the safety frame around the rollover axis before the impact is shown. In the course of an analysis, it was found that the angular velocity of the frame at the moment of the impact was 129 degrees per second (2.24 rad/s). The obtained values were used in the dynamic (impact) analysis of the frame as the boundary conditions of the model.

In the second stage, an analysis of the falling of the safety frame was carried out, striving to establish the permanent deformations on a simulation of a structure rollover test according to UNECE Regulation No.66/02.

As already mentioned, a surface geometric model of the frame based on a spatial model had been developed for the analysis of finite elements. In addition, in the development of this model, the frame tubular elements were correctly interconnected, eliminating the gaps between the frame elements for welding, which was necessary in the spatial model. The elements of the surface model were of different thicknesses: 1.5, 2, 3, and 4 mm. All elements were interconnected using bonded type component contact. In addition, a three-dimensional base was modeled at an angle of 18.9 degrees to the vertical plane of the safety frame. Thus, the computational model for impact analysis was constructed using mixed (“shell” and “solid” type) finite elements (base and H-beam “solid” type finite elements, the rest of the structure was “shell” type finite elements). “No penetration” type contact with a coefficient of friction of 0.6 was used for the contact between the frame and the base.

The analysis of the safety frame with an additional mass of 610 kg was performed using a true (computational) nonlinear steel S355 material model with kinematic hardening, as shown in Figure 7 below. The true stress–strain curve for simulation of S355 material was made according Al-Thairy et al. [21]. The mechanical properties of concrete were used to model the base material (Young’s modulus E = 23.0 × 10^9^ Pa, density 2400 kg/m^3^ and Poisson’s ratio 0.24).

The chosen geometry and the marginal conditions for the model conform to the requirements of the rollover test to the maximum possible extent. The angle of the model’s frame with a plane conforms to the angle where the structure becomes unstable (found in the course of the analysis of the falling). The initial angular velocity of the safety frame was chosen to be 2.24 rad/s, as established in the course of the above-mentioned analysis of the falling.

In Figure 8, the presented computational model is divided into shell type finite elements. In order to select the right grid size of the finite element, test calculations were performed with a partial frame model and the optimal size of the finite element was sought. The results of the calculations obtained by shredding a finite element grid were more accurate, but only up to a certain level. With a large increase in the number of finite elements, the computational error increases rapidly due to the approximate calculation method. In addition, as the size of the finite element decreases, the integration step decreases proportionally, resulting in a significant (non-linear) increase in the time to solve the problem. It consists of 37,249 nodal points and 31,782 finite elements.

In Figure 9, the summarized frame displacement fields 0.24 s after the beginning of the impact are shown; in Figure 10, fields of equivalent plastic deformations of the structure’s frame at the same time moment are shown. The summarized frame displacement fields make it possible to judge the suitability of the frame in the sense that the structural elements of the frame do not fall into the safety space, as shown in Figure 2. Even if the elements of the frame structure cross this safety space as little as possible, the safety frame structure would be considered unsuitable. Equivalent plastic deformations indicate how irreversibly the frame structure will be deformed. If the equivalent deformations exceed the critical values (for steel S355 ε > 0.20), the disintegration of the structure would start. The formulation of the problem would make it possible to see such a process of structural disintegration.

Figure 10 shows the equivalent plastic deformation fields of the structure’s frame, 0.24 s after the beginning of the impact. Furthermore, we can see that the largest deformations occur in the upper corners of the frame structure, as well as on the sides in the zone of maximum overall dimension. It is these local deformation zones that will determine the plastic deformations of the frame. In order to minimize the plastic deformations of the frame in these places, the structure should be stiffened.

### 3.2. A Natural Experiment of the Safety Frame Falling

A natural experiment of the safety frame falling was carried out, striving to establish the permanent deformations of the frame on the rollover test of the structure according to UNECE Regulation No. 66/02.

For the experiment, a platform that conforms to provisions of UNECE Regulation No. 66/02 was used. In Figure 11, a safety frame on a tip-up platform is shown. The initial position of the frame (the point where the experiment starts from) was chosen in a such way that the position of the frame is close to the one losing stability. 

The rollover of the safety frame was shot by a Phantom v711 high-speed camera.

The safety frame rollover experiment was fixed with the frequency of 1100 shots per second (1 shot: 909.09 μs).

In an analysis of the filmed materials, it was found that the frame was falling from the initial position until its corner touched the ground for about 1.66 s, and after four rebounds with decreasing amplitudes, became stable. In Figure 12 below, the filmed image of the safety frame 0.24 s after the beginning of the impact is shown. The number of frame rebounds allows us to judge the stiffness of the structure and the efficiency of shock absorption. In addition, this parameter is very valuable in verifying the computational model because the actual structure is welded (the frame has residual stresses) while the computational model is idealized.

In the course of the analysis of the data provided in the said materials, it was found that the maximum deformation of the frame took place 0.105 s after the beginning of the impact and it rebounded from the base 0.157 s after the beginning of the impact. The frame achieved the maximum amplitude of the first rebound 0.356 s after the beginning of the impact and reached the ground once more after 0.674 s.

After the rollover test, the frame was measured, and it was found that the maximum displacements of it were 132.46 mm. In Figure 13a, the silhouettes of the deformed frame and the frame before deformation obtained in course of the said measurement are provided. In Figure 13b, the deformed frame with safety space provided is shown. It is clear that the safety clearance was not compromised, and the frame was stiff enough to pass the test according to UNECE Regulation No. 66/02.

A numerical analysis of the results obtained from the safety structure rollover test and the data provided in the filmed materials of the above-mentioned natural experiment affirm that a good coincidence of the results was obtained, so the computational model should be considered correct. For example, in the natural test (the results were obtained from the filmed materials), the maximum deformation of the frame took place 0.103 s after the beginning of the impact and in the numerical experiment, the time was equal to 0.096 s; this is less than 7% difference. In the natural test, the frame rebounded from the base 0.157 s after the beginning of the impact and in the numerical experiment, the time was equal to 0.155 s; this is less than 1.2% difference. A good coincidence of the results of the calculations with the results of the natural experiment was confirmed by the permanent deformed contour of the safety frame. Figure 14 shows juxtaposed deformed forms of the safety frames, which were obtained during a natural experiment and numerical analysis. The maximum difference in results is less than 2%.

The contour of the safety frame prior to and after the deformation appeared during the rollover test (contour before deformation—black line) and the deformed state after the rollover (numerical analysis—orange line, experimental test—black dotted line). The results of the numerical analysis of the safety structure rollover test and experimental test shows a good coincidence of the results which are very similar.

### 3.3. Comparison of the Results of the Analytical Formula and the Finite Element Method

Since the height of the fall of the center of gravity of the safety frame was determined during the kinematic analysis to be 1050 mm, when the vehicle is in an initial unstable position with respect to the horizontal lower plane of the pit, the reference energy is:$$E_{R}=M\cdot g\cdot h_{1}=1140\;\times \;9.81\;\times \;1.05\;=\;11,743\;J.$$

The energy change of the safety frame structure frame shown in Figure 15 was obtained by performing a frame impact analysis. The maximum impact obtained at the impact of 10,178 J is slightly lower than the value obtained by empirical calculation (11,743 J), which is less than a 14% discrepancy.

## 4. Discussion

Upon applying numerical methods, a methodology for examining the structures by modeling a test for assessing their rollover crashworthiness according to the United Nations Economic Commission for Europe (UNECE) Regulation No. 66 “Uniform Technical Prescriptions Concerning the Approval of Large Passenger Vehicles with Regard to the Strength of Their Superstructure” has been developed.

The methodology was not developed according to the Federal Motor Vehicle Safety Standard 220 (FMVSS 220), but according to the (UNECE) Regulation No. 66. Our view is consistent with that of other authors that Regulation No. 66 of the United Nations Economic Commission for Europe (UNECE) is a more appropriate methodology because it assesses dynamic rather than static effects. In solving the static problem, the weight, acceleration, heeling angle, angular velocity, and other aspects necessary to obtain the most accurate result are not taken into account.

According to the above-mentioned methodology, an analysis of the bus safety structure frame’s rollover was carried out by numerical methods; in addition, the maximum and permanent deformations of the structure were established. It can be used to assess the safety and reliability of a structure. These set values are very important as they are directly related to passenger safety. The greater the deformation of the protection structure, the smaller the remaining safety space for passengers. Additionally, the smaller the remaining safety space for passengers, which decreases when the bus overturns, the more likely it is that passengers will be crushed, injured, or even killed.

In order to determine whether the safety frame model prepared and tested according to the developed methodology is suitable for use, a natural safety structure rollover test was carried out; the maximum and permanent deformations of the structure were established. The test was filmed using a high-speed camera Phantom v711 (with the frequency of 1100 shots per second (1 shot: 909.09 μs)). In addition, the relevant measurements were made.

The results of the numerical analysis of the safety structure rollover test and the data provided in the filmed materials of the above-mentioned natural experiment affirm that a good coincidence of the results was obtained, so the computational model should be considered correct. For example, in the natural test (the results were obtained from the filmed materials), the maximum deformation of the frame took place 0.103 s after the beginning of the impact and in the numerical experiment, the time was equal to 0.096 s. In the natural test, the frame rebounded from the base 0.157 s after the beginning of the impact and in the numerical experiment, the time was equal to 0.155 s.

After a completion of the above-described tests, it may be stated that sufficient safety of passengers and other traffic participants while using the vehicle is ensured. In addition, a duly formed model of a safety structure can be used for various simulations directly related to today’s conditions of exploitation, as well as the safety and reliability provisions.

This methodology could be used in the design of structures for bus safety frames and other vehicles. Using this methodology, manufacturers would save money for safety tests because the structure’s safety would be tested in the design stage.

In addition, according to the developed methodology, it is planned to solve another problem—the problem of optimizing the safety structure by minimizing the weight. In the structure of the safety frame, innovative materials would be used instead of steel, which would significantly reduce the weight of the structure while maintaining the existing strength properties.

## 5. Conclusions

In this work, according to the United Nations Economic Commission for Europe (UNECE) Regulation No. 66, the analysis of a spatial tubular structure of a bus safety frame was performed. Numerical simulations were performed using the Ansys software program. A natural experimental rollover test was performed in parallel with an identical bus safety frame structure. Based on the results of numerical modeling and the natural experiment, the conclusions are presented and the confirmation that the created numerical model is suitable and can be used in further work is accepted.

Using two different software packages for virtual bus safety frame analysis reduces the problem solution time by more than 6 times, while maintaining the same accuracy of results. In summary, we can say that the total computation time with the available computer equipment was 72 h, while using one software program (Ansys Autodyn 2019R1 version), the problem solution time would be more than 420 h. The modeling performed in this article allowed us to confirm the suitability of the methodology that will be applied to future simulations that will analyze a structure that is made of several completely different nonlinear materials—steel and fiberglass composite. Therefore, it was very important to develop a calculation methodology that would allow us to quickly and efficiently obtain results by analyzing different frame structures made of different materials.

## Figures and Tables

**Figure 1 sensors-20-07092-f001:**
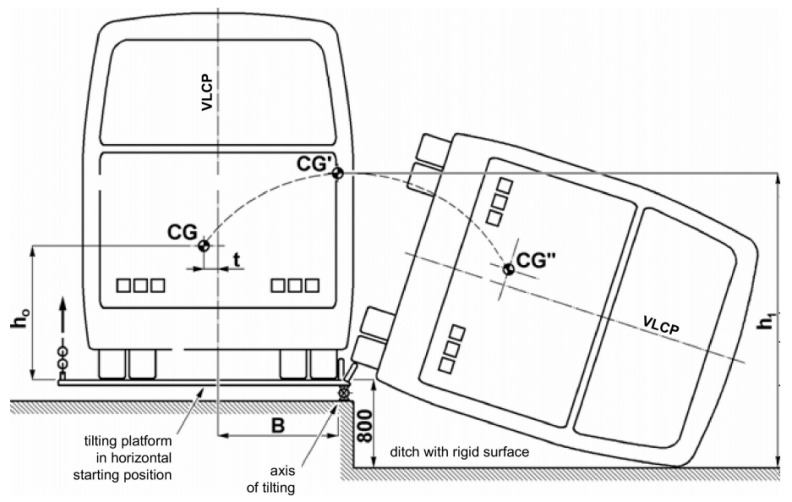
The computational model of a bus rollover [14].

**Figure 2 sensors-20-07092-f002:**
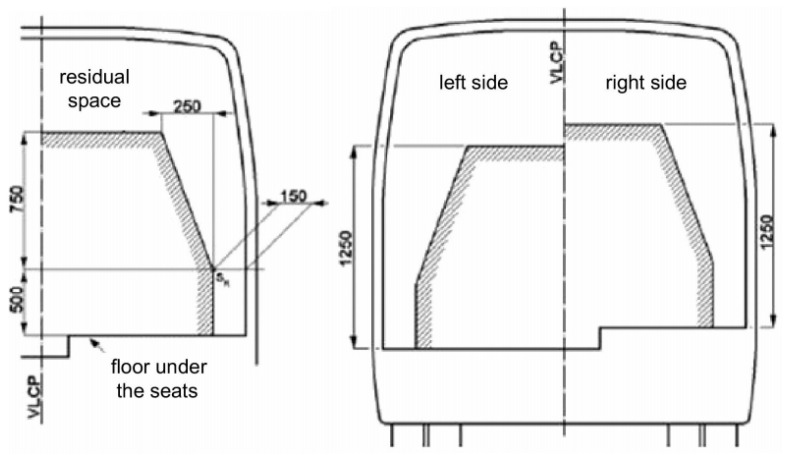
The model of the residual space layout [14].

**Figure 3 sensors-20-07092-f003:**
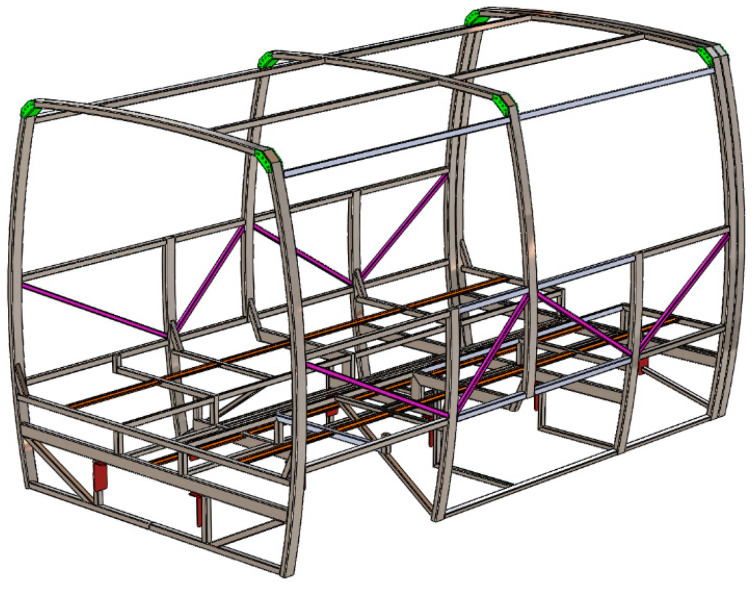
A geometrical model of the safety structure.

**Figure 4 sensors-20-07092-f004:**
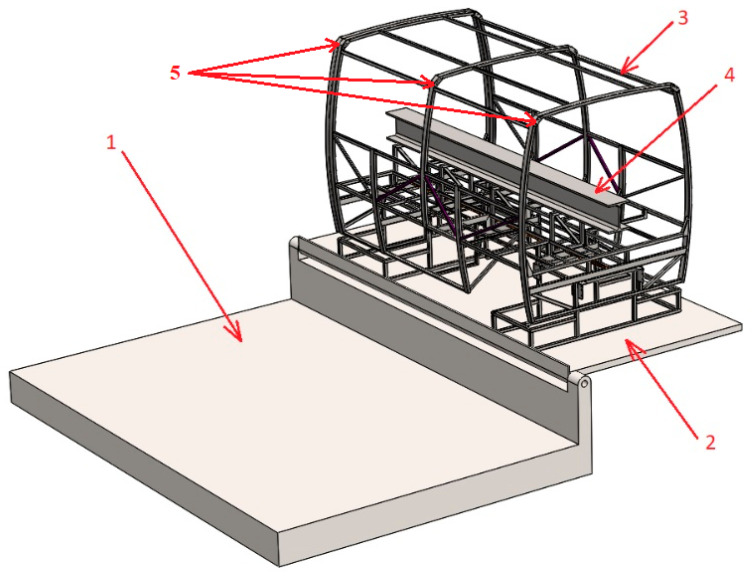
The computational model of a rollover of the safety frame: 1—concrete base, 2—tip-up platform, 3—rollover safety frame model, 4—additional weight, 5—sensors for contact detection.

**Figure 5 sensors-20-07092-f005:**
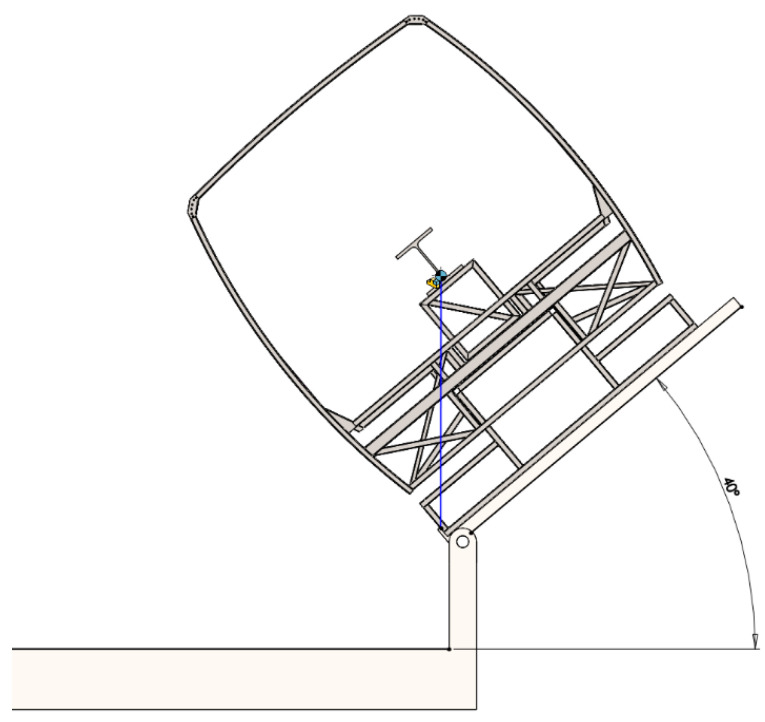
The position of the frame rollover when the frame is loaded with an additional mass of 610 kg.

**Figure 6 sensors-20-07092-f006:**
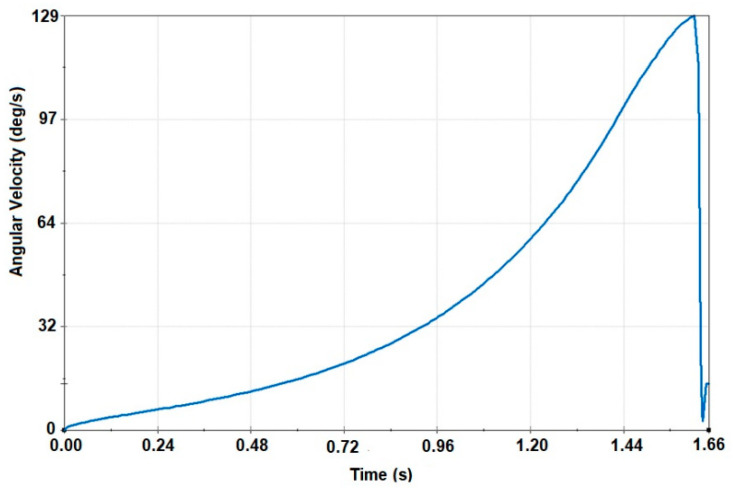
The alteration of the angular velocity of the safety frame around the rollover axis before the impact.

**Figure 7 sensors-20-07092-f007:**
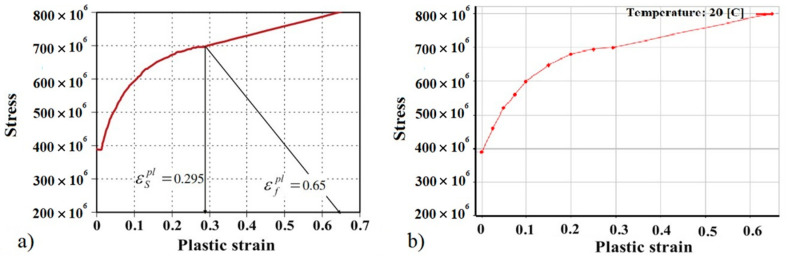
True stress–strain curve used to simulate S355 material: (**a**) Al-Thairy et al. [21], (**b**) used in finite element code.

**Figure 8 sensors-20-07092-f008:**
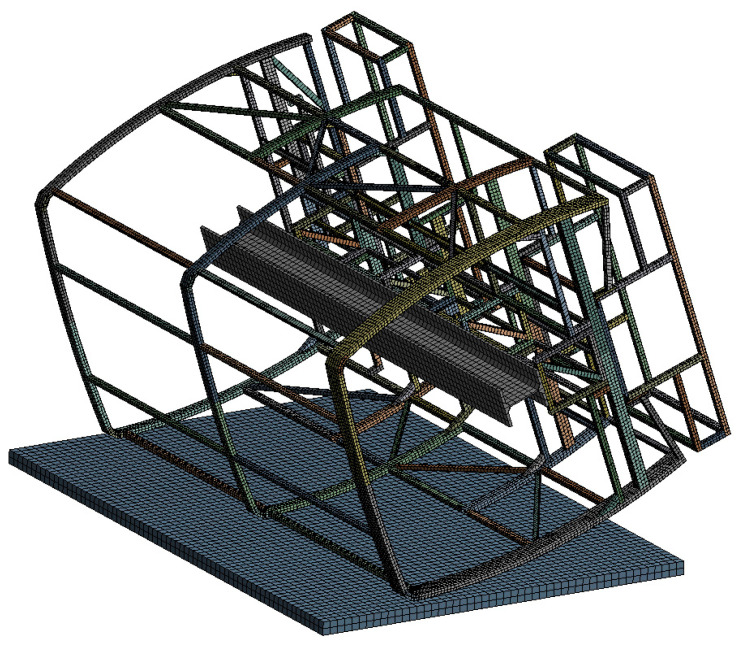
A finite element model of the safety structure.

**Figure 9 sensors-20-07092-f009:**
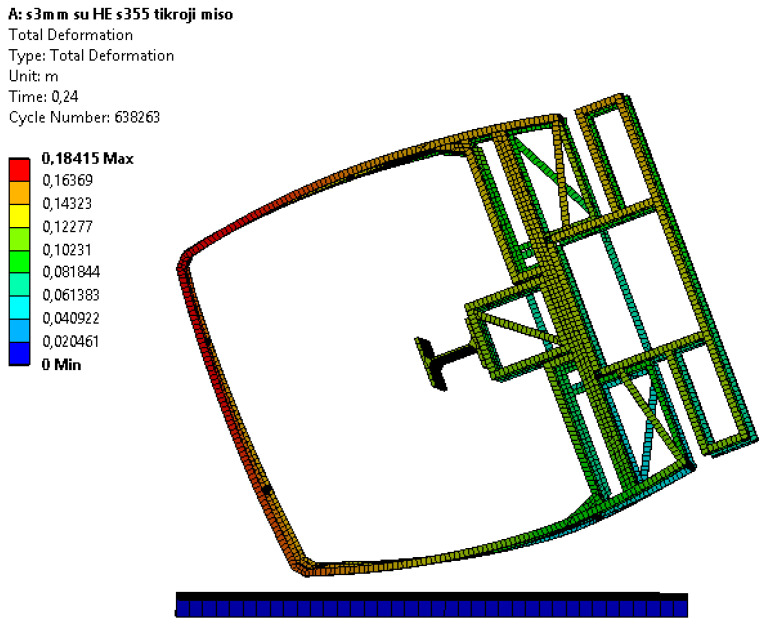
The frame displacement fields 0.24 s after the beginning of the impact, mm.

**Figure 10 sensors-20-07092-f010:**
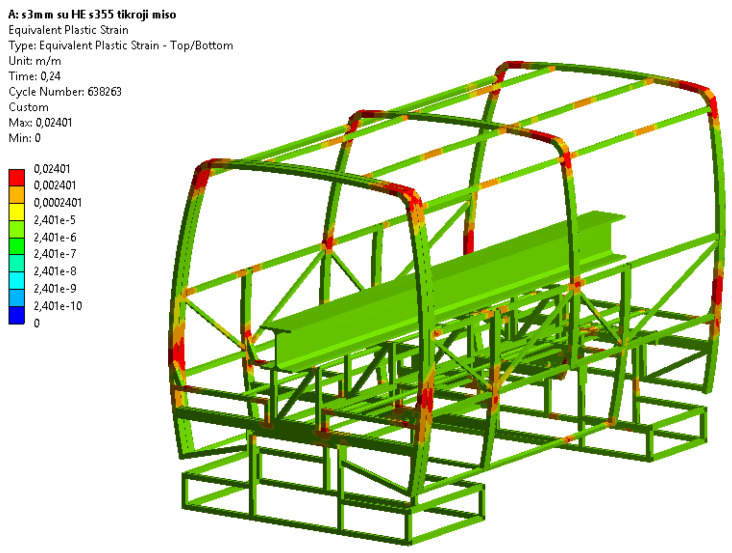
Fields of equivalent plastic deformations of the structure’s frame 0.24 s after the beginning of the impact, mm/mm.

**Figure 11 sensors-20-07092-f011:**
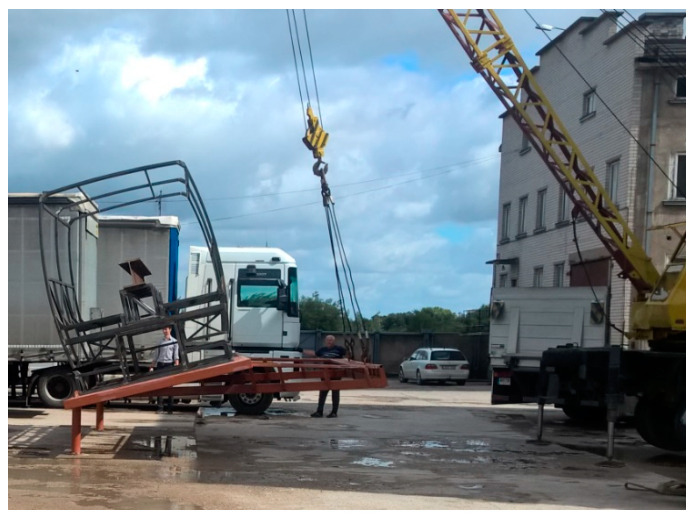
The safety frame before the natural rollover experiment.

**Figure 12 sensors-20-07092-f012:**
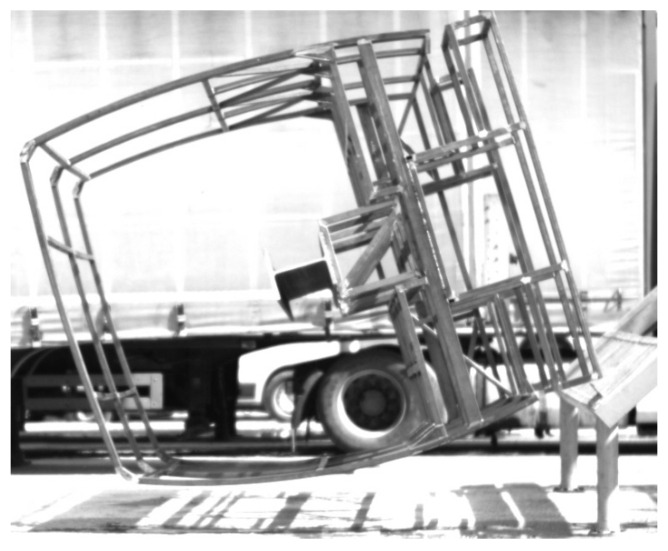
The safety frame 0.24 s after the beginning of the impact.

**Figure 13 sensors-20-07092-f013:**
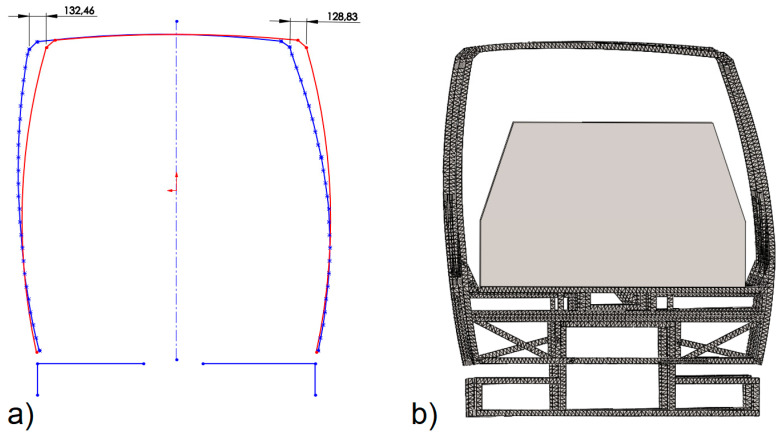
The contours of the safety frame: (**a**) the frame contour before and the frame contour after the deformation appeared during the rollover test (results of the measurements), (**b**) deformed frame with safety space.

**Figure 14 sensors-20-07092-f014:**
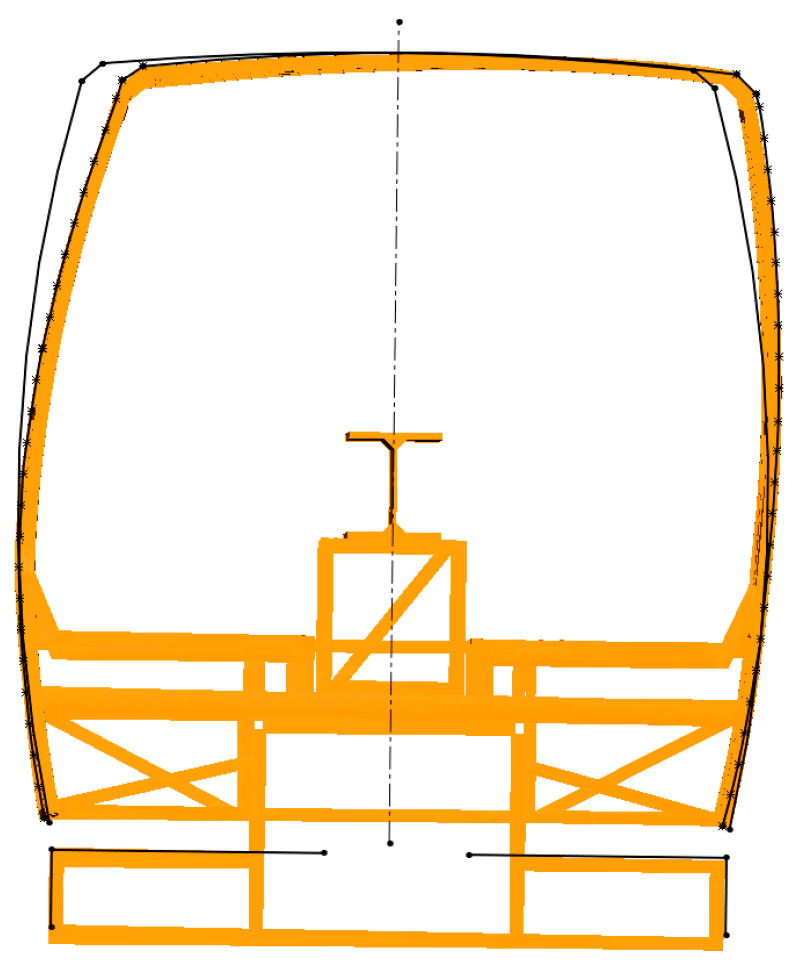
The contour of the safety frame prior to and after the deformation.

**Figure 15 sensors-20-07092-f015:**
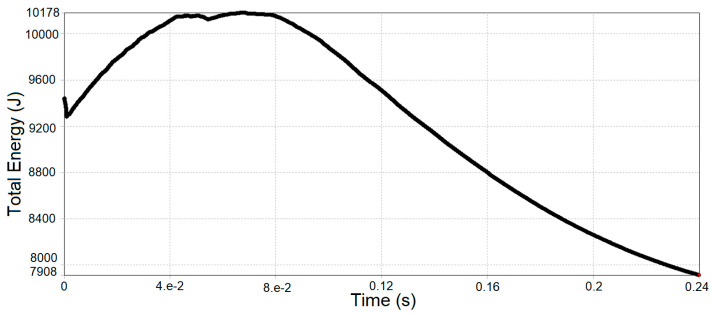
Variation of the total energy of the bus safety frame over time, J.

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
