# Peer review of "An Analysis of the Reliability of a Bus Safety Structure on Carrying Out the Numerical and Experimental Tests"

_sensors, 2020, doi:10.3390/s20247092_

Round 1

Reviewer 1 Report

Dear Editor, Dear Authors,

In this paper, a comparison between experimental testing and numerical simulation is presented for bus safety structure according to UNESCE regulation No. 66. The numerical simulation is divided on two parts performed in different software. I do not know if Autodyn allows transition between deformable and rigid so the analysis can be performed using one software (just to notice that I have nothing against using two separate software). The results from the first (rigid body simulation) were used as an initial condition for the final finite element analysis. The language is fluent and easy to read. However, I have a few questions.

About material model…At row 232…what is “REAL” nonlinear model?
The material model was taken from the literature?
Are the model parameters for the same plate thickness as the bus frame?
Is it only one plate thickness used? If not do you think that plate thickness influences the material response?
What about strain rate hardening?
As far as I can see on your figure 10 the max plastic strain is 0,024 which is 2,5% …This locate your stress just at the beginning of the curve shown at figure 7. which buy the way at that part is represented as a flat curve if I'm seeing correctly? This is than the bilinear model without strain hardening in this region of the curve. How will you comment on this?

And the famous question…What is the novelty of this paper? What is new or valuable in this paper that I can take as a core message? In terms of….when you do a numerical simulation…use this material model…extract the material parameters of the true stress-strain curve by this procedure….use this size of the mesh. check the energies...

Can you please also comment on how the content of this journal is connected with the aim and scope of Sensors?

Kind regards and everything best.

Author Response

Dear reviewer,

Thanks for Your comments. They were really helpful.
In response to Your comments, we supplemented our article and made it clearer.

Provided answers to your questions. Hopefully they are detailed enough.

Thank you

Sincerely,
Authors

Reviewer 2 Report

This manuscript proposed a methodology for assessing bus rollover crashworthiness applying numerical methods. The methodology proposes the use of kinematic simulation and finite element analysis to reproduce the rollover test according to UNECE Regulation No. 66. In accordance with the proposed methodology, a virtual test of the safety structure of a bus was carried out. The mathematical model results were evaluated using the results of a real test.

The methodology is well known and does not represent an advance in the state of the art. It has been used by manufacturers and designers because virtual testing is an accepted method for approval.

Computer simulation of rollover test as an equivalent approval method is one of the annex of the Regulation No. 66. The methodology, between lines 31-155, is a summary of this regulation.

The need to use kinematic simulation has not been adequately demonstrated. Furthermore, the kinematic model has not been described in detail, for example, how the contacts, joints, etc. have been defined.

In the same way, the finite element model has not been described in detail.

Finally, the correlation of the model has been poorly demonstrated. I recommend checking methods like RSVVP and CORA

Author Response

(The authors gave the same response as above.)

Round 2

Reviewer 1 Report

Dear authors,

Thank you for your answers to my questions. However, I have a few doubts about your answers

Q1.

You are saying “However, the steel properties of different thickness are the same for all elements because the material of these elements is steel of the same brand“. I'm quite sure that this is not true. However, I accept this kind of answer.

Q2.  Please take a look once again, I think in your answer you mixed strain values of 2.5% with 25%.

Q3.  I would also like to please you to think about some more valuable originality of your paper and meaningful message to your future readers. 

Author Response

Dear reviewer,

Thank you for your questions and attention.

It is a great pleasure for You to accept our answers.

We  hope the answers to Your questions are informative enough.

Sincerely,

Authors

Reviewer 2 Report

For future work I suggest improving the correlation of the model. For example, measuring the acceleration of different points or measuring force vs deformation in different points of the frame. I suggest to review the methodologies used to objective evaluation of time-history signals (ISO 18571, CORA)

Author Response

Dear reviewer,

Thank You for Your suggestion. We value it very much because we understand that the correlation of results, their analysis is very important.

In the future works we will use Your suggested method.

Sincerely,

Authors
